# The Biological Characteristics and Differential Expression Patterns of *TSSK1B* Gene in Yak and Its Infertile Hybrid Offspring

**DOI:** 10.3390/ani13020320

**Published:** 2023-01-16

**Authors:** Yanjin Zhu, Bangting Pan, Xixi Fei, Yulei Hu, Manzhen Yang, Hailing Yu, Jian Li, Xianrong Xiong

**Affiliations:** 1Key Laboratory of Qinghai-Tibetan Plateau Animal Genetic Reservation and Exploitation of Ministry of Education, Southwest Minzu University, Chengdu 610041, China; 2Key Laboratory of Animal Science of National Ethnic Affairs Commission, Southwest Minzu University, Chengdu 610041, China

**Keywords:** yak, cattle–yak, *TSSK1B*, expression, sterility

## Abstract

**Simple Summary:**

The *TSSK1B* gene has been demonstrated to play pivotal roles during spermatogenesis and is associated with male fertility. However, the potential mechanism on whether and how *TSSK1B* disrupts the fertility of yaks remains unknown. In this study, *TSSK1B* was found to be specifically expressed in the testes of yaks and especially highly expressed in adults. In contrast, it was rarely expressed in the testes of male cattle–yak, the hybrid F1 generation of the yak. The promoter region of *TSSK1B* in the adult cattle–yak testis was hypermethylated compared with that in the yak, which may be related to male cattle–yak infertility. This study provides the basis for further study of yak reproduction and male cattle–yak sterility.

**Abstract:**

This study aimed to investigate the spatially and temporally expressed patterns and biological characteristics of *TSSK1B* in male yaks and explore the potential correlation between *TSSK1B* and male sterility of the yak hybrid offspring (termed cattle–yak). First, the coding sequence (CDS) of *TSSK1B* was cloned by RT-PCR, and bioinformatics analysis was conducted with relevant software. Quantitative real-time PCR (RT-qPCR) was employed to detect the expression profile of *TSSK1B* in various tissues of male adult yaks, the spatiotemporal expression of *TSSK1B* in different stages of yak testes, and the differential expression of *TSSK1B* between yak and cattle–yak testes. The cellular localization of *TSSK1B* was determined by immunohistochemistry (IHC). Furthermore, the methylation status of the *TSSK1B* promoter region was analyzed by bisulfite-sequencing PCR (BSP). The results showed that *TSSK1B* was 1235 bp long, including 1104 bp of the CDS region, which encoded 367 amino acids. It was a conserved gene sharing the highest homology with *Bos mutus* (99.67%). In addition, the bioinformatics analysis revealed that TSSK1B was an unstable hydrophilic protein mainly containing the alpha helix of 34.06% and a random coil of 44.41%, with a transmembrane structure of 29 amino acids long. The RT-qPCR results demonstrated that *TSSK1B* was specifically expressed in yak testes compared with that in other tissues and especially highly expressed in adult yak testes. On the contrary, *TSSK1B* was hardly expressed in the testis of adult cattle–yak. IHC confirmed that *TSSK1B* protein was more strongly expressed in the testes of adult yaks than in their fetal and juvenile counterparts. Interestingly, nearly no expression was observed in the testes of cattle–yak compared with the corresponding testes of yak. Bisulfite-sequencing PCR (BSP) revealed that the methylated CpG sites in the *TSSK1B* promoter region of cattle–yak was significantly higher than that in the yak. Taken together, this study revealed that *TSSK1B* was specifically expressed in yak testes and highly expressed upon sexual maturity. Moreover, the rare expression in cattle–yak may be related to the hypermethylation of the promoter region, thereby providing a basis for further studies on the regulatory mechanism of *TSSK1B* in male cattle–yak sterility.

## 1. Introduction

The yak (*Bos grunniens*), which mainly lives in the Qinghai–Tibetan Plateau and its vicinity 3500 m above sea level altitude, has strong adaptability to the alpine and hypoxic environment [1]. As one of the specific species that could subsist and reproduce in that area, the yak plays a crucial role in local livelihood and economy. However, the growth and reproductive performance of the yak are quite deficient due to the adverse environment of the plateau. To ameliorate this situation and improve the economic performance of the yak, different kinds of cattle with outstanding traits were introduced to hybridize with the yak, and the F1 generation was termed the cattle–yak [2]. The growth performance, nutrient digestibility, meat quality, and milk performance of the cattle–yak are significantly enhanced, which is called heterosis [3,4,5]. Simultaneously, its adaptability to the Qinghai–Tibetan Plateau is equivalent to that of the yak. Therefore, cattle–yak breeding is an important method to improve the production efficiency of the yak, which is beneficial for the development and economy in that region. However, the male cattle–yak is infertile, as the spermatogenesis is blocked, which greatly restricts its breeding capability and the effective utilization of heterosis. In recent decades, advances in the potential mechanisms have been demonstrated on the basis of cytobiology [6], transcriptomes [7], and proteomics [8] analysis, but uncovering the mechanism of male cattle–yak sterility remains a tremendous challenge.

Testis-specific serine threonine kinase 1B (*TSSK1B*), a kind of protein kinase, plays a key role in numerous biological processes, such as gene expression, signal transduction, and cellular metabolism, by transferring the phosphate groups to the serine, threonine, or tyrosine residuals of substrates. These kinds of phosphorylation of proteins are reversible post-translational modification, and they regulate the functions of cells [9]. Five members of *TSSKs*, namely *TSSK1*, *TSSK2*, *TSSK3*, *TSSK4*, and *TSSK6*, were found in previous studies [10,11,12,13,14,15] to be specifically expressed in the spermatozoa and testes, indicating that *TSSKs* play pivotal roles in male reproduction. In humans, *TSSK1* presents as *TSSK1A*, a pseudogene that links to *TSSK2* to form a tandem [16]. In the targeted knockout of *TSSK1* and *TSSK2*, male chimeras appeared, and germ cells with mutant allele were generated, indicating that *TSSKs* are related to the infertility of male chimeras [17]. Other research also found that *PPP1CC2* could form the complex with *TSSK1* by directly interacting with testis-specific serine kinase substrate [18]. A recent transcriptome study reported that *TSSK1B*, *TSSK2*, and *TSSK6* were significantly downregulated in cattle–yak testes compared with those in the yak [19], indicating that *TSSKs* may be associated with the blocked spermatogenesis of cattle–yaks. Hence, understanding the role of *TSSK*s in male spermatogenesis is fundamental for extensive research on the mechanism of male infertility.

This study aimed to characterize yak *TSSK1B* by providing the complete coding region sequence and molecular characterization. Tissue-specific *TSSK1B* mRNA expression levels in several yak tissues were detected. The spatial and temporal expression of *TSSK1B* in yak testes among various stages were determined, along with the comparison of the relative *TSSK1B* mRNA expression levels in testis tissue between yak and cattle–yak through quantitative real-time PCR (RT-qPCR). In addition, immunohistochemistry (IHC) was performed as a supplementary analysis of *TSSK1B* expression in the testis development of yaks and cattle–yaks. The methylation status of the *TSSK1B* promoter region was determined by bisulfite-sequencing PCR (BSP) to explore the possible cause that influenced the expression status of *TSSK1B* in cattle–yak testis. This study could facilitate the understanding of the biological role of *TSSK1B* in male reproduction and help explore the potential mechanism of cattle–yak male sterility.

## 2. Materials and Method

### 2.1. Ethics Statement

All chemicals used were purchased from Sigma Chemical Company (St. Louis, MO, USA), unless otherwise noted. All animal-related experiments were in accordance with the Southwest Minzu University Animal Care and Use Committee and performed in accordance with the National Research Council’s Guide Principles for Animal Welfare and Ethics.

### 2.2. Tissues Collection

The heart, liver, spleen, lung, kidney, intestine, muscle, and testis tissue samples of three healthy adult Maiwa yaks and hybridized cattle–yaks (3–4 years old) and testis tissue samples of three fetal (5–6 months old) and juvenile (1–2 years old) yaks were collected from a slaughterhouse in Hongyuan County (altitude of approximately 4000 m), Sichuan Province. The collected samples were conserved in a liquid nitrogen container immediately on the spot after cleaning with PBS for subsequent experiments.

### 2.3. RNA Extraction and cDNA Synthesis

The total RNAs of the collected tissues were extracted with TRIzol reagent in accordance with the manufacturer’s instructions. The concentration and purity of the RNA samples were detected by a NanoDrop 2000 spectrophotometer (ThermoFisher, Waltham, MA, USA). The ReverAid First Strand cDNA Synthesis Kit (ThermoFisher, Waltham, MA, USA) was used to reverse the transcription of RNA, following the manufacturer’s guidelines. The quality of the cDNA samples was evaluated by NanoPhotometer N60 (Implen, Westlake Village, CA, USA) and then stored at −20 °C for further use.

### 2.4. Gene Cloning and Sequencing

The primers of *TSSK1B* (shown in Table 1) were designed by NCBI Primer-BLAST (NCBI, Bethesda, MD, USA) according to the predicted sequence of hybrid cattle (Accession number: XM_027566040.1). The previous cloning protocol of the laboratory [20] was referred in each amplification system (25 μL), including 1 μL of forward and reverse primers, 12.5 μL of 2×Rapid Taq Master Mix (Vazyme, Nanjing, China), 1 μL of cDNA synthesized from all tissue samples, and 9.5 μL ddH_2_O. The PCR procedure was as follows: 95 °C for 3 min, followed by 35 cycles of 95 °C for 15 s, 57 °C for 15 s, 72 °C for 1 min, and 72 °C for 5 min. The PCR products were identified by gel electrophoresis. The purified amplification products were then recovered and ligated to pMD19-T vector (Takara, Japan) and finally transformed into *E. coli* DH5a cells. The positive colonies of *TSSK1B* were selected for sequencing by Sangon Biotech Co., Ltd. (Shanghai, China).

### 2.5. Bioinformatics Analysis

The yak *TSSK1B* cDNA sequences were subjected to BLAST analysis for confirmation. Then the cDNA sequence was analyzed by the ORF Reader program of NCBI (https://www.ncbi.nlm.nih.gov/orffinder/ (accessed on 15 August 2022). The amino acid sequence was predicted by DNAMAN 9.01, as described in a previous study [21], and the properties of the predicted amino acid sequence were inspected by BioEdit 7.0 software, the online Expasy ProtParam Tool (https://web.expasy.org/protparam/ (accessed on 15 August 2022)), and the SignalP-5.0 online program (https://services.healthtech.dtu.dk/service.php?SignalP-5.0) (accessed on 15 August 2022). In order to predict the secondary and tertiary structures and the domains of proteins encoding *TSSK1B*, the online tool SOPMA portal (https://npsa-prabi.ibcp.fr/cgi-bin/npsa_automat.pl?page=npsa_sopma.html (accessed on 15 August 2022)), the SWISS-Model (https://swissmodel.expasy.org/interactive) (accessed on 15 August 2022), and InterPro (http://www.ebi.ac.uk/interpro/ (accessed on 15 August 2022)) were employed, respectively [22]. The phosphorylation sites were also analyzed by the online portal NetPhos 3.1 (https://services.healthtech.dtu.dk/service.php?NetPhos-3.1) (accessed on 15 August 2022). The Nucleotide BLAST portal of NCBI (https://blast.ncbi.nlm.nih.gov/Blast.cgi (accessed on 15 August 2022) was utilized to obtain the homeotic sequence from various species based on the NCBI GenBank database, following the alignment of the coding sequence (CDS) region of cloning *TSSK1B* sequence with the retrieved homeotic sequences. An evolutionary tree was established by MEGA 7.0 software in accordance with the abovementioned analysis.

### 2.6. Quantitative Real-Time PCR (RT-qPCR)

The relative mRNA expression of *TSSK1B* in each tissue sample of yak and cattle–yak was detected by RT-qPCR, following the previous study [20]. The primers of *TSSK1B* for RT-qPCR were designed, as shown in Table 1. Each amplification system (20 μL) included 1 μL of forward and reverse primers, 10 μL of 2×NovoStart^®^SYBR qPCR SuperMix Plus (Novoprotein, Suzhou, China), and 1 μL of diluted cDNA supplied with 7 μL RNase free water up to 20 μL. The CFX96^TM^ system (Bio-Rad, Hercules, CA, USA) was used with the following program steps: 95 °C for 1 min, 40 cycles of 95 °C for 20 s, and 60 °C for 1 min, followed by dissociation curve and cool down. The relative fold change of genes was calculated by 2^–∆∆Ct^ method, and glyceraldehyde-3-phosphate dehydrogenase (*Gapdh*) was used as the housekeeping gene for data normalization. Each sample analysis was repeated independently in triplicate. For easy comparison of the relative expression of *TSSK1B* in various organs, the expression in heart was set to 1 (control group). The *TSSK1B* expression of adult yak testis was also set as the control group when contrasting the differential expression in yak testes of different ages and the expression in adult cattle–yak testes.

### 2.7. Immunohistochemistry (IHC) Analysis

IHC was performed as described in the previous study [23]. In brief, testicular samples from fetal, juvenile, and adult yaks and adult cattle–yaks (n = 3, respectively) were collected and immediately fixed with 4% paraformaldehyde. After being embedded in paraffin, 4 μm sections were obtained, and then the sections were deparaffinized and rehydrated in ethanol-to-water graded series, followed by hydrating with an ethanol gradient from 100% to 70%. After antigen retrieval was performed with 5% FBS at room temperature for 1 h, the sections were incubated with primary *TSSK1B* antibody (1:50, PA5-55706, Thermo, Waltham, MA, USA) overnight at 4 °C, washed with PBS, and incubated with HRP-conjugated secondary antibody (1:200, bs-0295G-HRP, Bioss, Beijing, China) at 37 °C for 2 h. Images were obtained by using a Zeiss LSM800 confocal microscope (Observer Z1, Zeiss, Germany).

### 2.8. Genome DNA Extraction and Bisulfite Modification

Samples of less than 25 mg were incubated overnight at 55 °C with protein kinase K and RNase before the utilization of DNA Extraction Kit (Solarbio, Beijing, China) Kit, in accordance with the manufacturer’s instructions, to extract the genome DNA from the testis tissues of yaks and cattle–yaks. The purified DNA samples were treated with the EpiArt^®^ DNA Methylation Bisulfite Kit (EM101, Vazyme, Nanjing, China) by following the instructions of the manufacturer. The bisulfite treatment converted the unmethylated cytosine bases to thymine bases (C-T), whereas the methylated cytosine bases remained unchanged [24].

### 2.9. CpG Islands (CGI) Prediction and Bisulfite-Sequencing PCR (BSP)

For the analysis of the *TSSK1B* promoter methylation status of yak and cattle–yak testis, a 2000 bp sequence before transcription start sites of *TSSK1B* was selected to predict the CGIs [25]. The specific primers for BSP were designed in accordance with the online program MethPrimer (http://www.urogene.org/methprimer/ (accessed on 9 September 2022), as shown in Table 1).

BSP was performed by using 2× EpiArt ^®^ HS Taq Master Mix (EM202, Vazyme, Nanjing, China), following the instructions of the manufacturer. The reaction conditions were as follows: 95 °C for 5 min, followed by 45 cycles of 95 °C for 30 s, 51 °C for 30 s, 72 °C for 30 s, and 72 °C for 5 min. The bisulfite-converted genomic DNA was used as the template. The PCR products were identified on 2.5% gel electrophoresis to purify the PCR products. Ten replicated amplification products of yak and cattle–yak were randomly selected and delivered to Sangon Biotech Co., Ltd. (Shanghai, China), for purification and sequencing. The sequencing results were aligned and analyzed by the BiQ-analyzer software. The methylation levels were represented as the ratio of unchanged CpG sites to all CpG sites in each sample.

### 2.10. Statistical Analysis

The mRNA expression and methylation status of *TSSK1B* were analyzed by SPSS 20.0, with one-way analysis of variance (ANOVA), and histograms were depicted by GraphPad Prism 8.0. The data are presented as the mean ± standard error (SE). A *p* < 0.05 was considered significantly different. * Represents *p* < 0.05, and ** represents *p* < 0.01.

## 3. Results

### 3.1. TSSK1B cDNA Cloning and Molecular Characterization

The cDNA from the testis tissue of yak was employed as the cloning template. According to the sequencing results, a 1235 bp fragment was obtained from yak testis (Figure 1A). The yak *TSSK1B* comprised a 1104 bp CDS region, which was predicted to encode 367 amino acids (Figure 1B). Aligning the sequence of yak *TSSK1B* with the orthologs derived from the NCBI showed the highest similarity of 99.67% to *Bos mutus*, followed by 99.42%, 98.27%, 96.29%, 94.97%, 94.72%, 89.99%, 88.56%, 88.35%, and 86.48% to *Bos taurus*, *Bubalus bubalis*, *Cervus elaphus*, *Capra hircus*, *Ovis aries*, *Equus asinus*, *Camelus ferus*, *Mustela erminea*, and *Marmota monax*, respectively. This finding indicated that the *TSSK1B* of yak was similar to that of cattle and highly conserved during evolution. A phylogenetic tree was also constructed and showed that the *TSSK1B* of yak was found to be clustered in an isoform with *B. mutus*, consistent with the ortholog analysis (Figure 2).

The deduced amino acid sequence was employed to analyze the protein structure and properties of yak *TSSK1B*. The predicted molecular weight was 41.5 kDa, with an isoelectric point (pI) of 7.49. The secondary structure of *TSSK1B* was mainly composed of 34.06% alpha helix, 6.81% beta turn, 14.71% extended strand, and 44.41% random coil (Figure 3A). The predicted tertiary structure of *TSSK1B* protein was close to the secondary structure (Figure 3B), and the characteristic sequence of protein kinase was obtained as the online report suggested. The estimated aliphatic index was 81.34, and the hydropathicity value was −0.532, which indicated that *TSSK1B* was a hydrophilic protein (Figure 3C). According to the signal peptide and transmembrane segment detection, a transmembrane structure of 29 amino acids long from 197 to 225 locus of the peptide chain was acquired (Figure 3D), whereas no signal peptide was discovered. The phosphorylation site analysis found twenty-nine positive phosphorylation sites, including nineteen serines, seven threonines, and three tyrosines, in yak *TSSK1B* (Figure 3E). The instability index was 57.06 (>40), implying that this protein was unstable. The prediction of the *TSSK1B* protein domain suggested that it possessed protein kinase activity and an ATP binding site that were usually considered to be the portions of the protein phosphorylation process.

### 3.2. Expression Pattern of TSSK1B in Yak and Cattle–Yak Tissue

RT-PCR and RT-qPCR were utilized to explore the expression pattern of *TSSK1B* in the testes of yak and cattle–yak, and the amplification products with a similar template concentration were detected by agarose gel electrophoresis. The expression levels of *TSSK1B* in eight tissues of adult yak (heart, liver, spleen, lung, kidney, intestine, muscle, and testis) were determined by RT-qPCR. The results indicated that the transcription level of *TSSK1B* in testis was significantly higher (*p* < 0.01) than that in the other seven tissues (Figure 4A), and no significant difference was found among these seven tissues. This finding is consistent with the gel electrophoresis image, in which a unique band appeared in the testicular sample (Figure 4D). The comparison of the mRNA expression of *TSSK1B* in yak testis from different ages implied that *TSSK1B* was highly expressed in adult yak testes and extremely lowly expressed in the fetal and juvenile stages (*p* < 0.01, Figure 4B), which was also identified by the absent band in the gel electrophoresis of fetal and juvenile yak testis samples (Figure 4E). A notable detail was that the RT-PCR and RT-qPCR reflected that *TSSK1B* was highly expressed in male adult yak testes and almost not expressed in cattle–yak testes (*p* < 0.01, Figure 4C, F).

### 3.3. The Immunohistochemistry Analysis of Yak and Cattle–Yak Testes

The cellular localization and differential expression of *TSSK1B* were evaluated by IHC (Figure 5), and the results indicated that *TSSK1B* was mainly expressed in the testes of adult yaks, including spermatogonial stem cells, spermatogonium (SP), primary spermatocyte (PS), secondary spermatocyte, and surrounding Leydig cells. Low expression was also found in the fetal and juvenile yak testes. However, almost no expression was observed during the growth of cattle–yak testes, significantly different from yaks. Compared with yak testes, the morphology of seminiferous tubule in cattle–yak testes was not significantly different at the fetal stage, but the cavity was smaller in adult cattle–yak testes, and only low numbers of SP, PS, and Sertoli cells (SCs) were loosely distributed in the cattle–yak testis.

### 3.4. DNA Methylation Status in the Promoter Region of TSSK1B

In accordance with BSP and sequencing, the DNA methylation status in the *TSSK1B* promoter region of yak and cattle–yak testes was determined. The agarose gel electrophoresis image (Figure 6A,B) indicated that the bisulfite-converted DNA fragments with 231 bp were obtained from yak and cattle–yak testes. A total of 15 available CpG sites of the amplified products were selected to analyze the methylation status (Figure 6C,D). The methylation frequencies of these 15 CpG sites in the yak and cattle–yak testis samples were 22.0% (33/150) and 45.3% (68/150), respectively (Figure 6E). These results suggested that the promotor region of *TSSK1B* was hypermethylated in the cattle–yak testis (*p* < 0.01).

## 4. Discussion

As a member of the protein kinase family, *TSSK1B* plays a key role in the germ-cell development of male animals. This study was the first to explore the molecular characterization and expression profile of yak *TSSK1B*, and the results indicated that *TSSK1B* was a highly conserved gene with a significant spatiotemporal-specific expression pattern. Furthermore, the mRNA and protein expression levels of *TSSK1B* was highly expressed in adult yak testis, accompanied by hypomethylation in the promotor region, compared with that of cattle–yak. This study may be beneficial for improving the male reproductive efficiency of yaks and exploring the mechanism of male sterility in the cattle–yak.

First, the completed cDNA sequence of *TSSK1B* was obtained from adult yak testis, and the deduced CDS region encoded 367 amino acids, similar to the results of the previous study on other species [26,27,28]. The ortholog analysis results suggested that *TSSK1B* was highly conserved during evolution among mammals, and the homology analysis results showed that the *TSSK1B* in yaks had similar homology to that in *B. mutus* (wild yak) and *B. taurus* (cattle), which was higher than 98%. Although the lowest homology was found in *M. monax,* it was also more than 86%. The results of the present study are consistent with those of a previous study on the Pacific abalone [29]. Furthermore, working as a protein kinase, *TSSK1B* possessed a specific catalytic domain of protein kinase, and it probably catalyzed the transfer of the gamma-phosphory l group from ATP to hydroxyl groups in specific substrates, such as serine or threonine residues of proteins. Similar to other types of protein kinase, *TSSK1B* plays a crucial role in intercellular signaling [9]. The findings on multiple phosphorylation sites, especially those enriching serine and threonine residues, confirmed this feature. The property analysis of *TSSK1B* protein showed instability, which may be related to its potential catalysis function during the phosphorylation and dephosphorylation process. These results indicated that the yak *TSSK1B* was highly conserved in evolution, with poor protein stability.

The testis is a crucial organ for male domestic animals to generate germ cells and sexual hormones. In the testis, the spermatogenesis could be divided into three phases: the proliferation stage, where the spermatogonia generates a pool of spermatocytes by mitotic division; the meiosis stage, where haploid spermatids are generated; and the spermiogenesis stage, which allows the spermatids to be differentiated into spermatozoa [30]. In the current study, the expression pattern of *TSSK1B* in various yak tissues was detected by RT-qPCR, and the results showed that *TSSK1B* was specifically expressed in yak testis, which was typically observed in mammalian *TSSK1B* and in agreement with other previous study [13,31]. Furthermore, the mRNA expression abundances of yak *TSSK1B* transcripts were correlated with the age of the males, in which a positive relationship could be observed between *TSSK1*-like expression levels and the degree of testis maturation. Based on the result of the RT-qPCR, *TSSK1B* was highly expressed in adult yak testis and significantly differed with the stage before sexual maturity. The IHC data were in agreement with the results of the RT-qPCR and previous studies [32]. In addition, the expression of *TSSK1B* mRNA in yak testis was significantly higher than that in their cattle–yak counterparts. This finding indirectly provided a possible assumption of *TSSK1B* involvement in male germ-cell differentiation during late-phase spermatogenesis. The previous study showed that the *TSSK* family was almost highly expressed in the testis compared to that in other tissues in the mouse, and the expression started at 21 d after birth, indicating that *TSSKs* were expressed after meiosis [15]. Furthermore, *TSSK1* was expressed in the later stage of spermiogenesis and mainly detected in condensed spermatids [15]. However, additional monitoring for *TSSK1B* expression in older males may be required to confirm the present study’s hypothesis and verify the developmental stage showing the onset of *TSSK1B* expression during spermatogenesis.

As the F1 hybridized offspring of yak with other cattle, cattle–yaks attract much attention due to their remarkable traits compared with yak, though the male cattle–yak is infertile. The expression of *TSSK1B* in male cattle–yak testis was investigated in this study to explore the potential reason for sterility. A notable detail is that the relative expression of *TSSK1B* in adult cattle–yak testis was significantly lower than that in yak testis, indicating that the male sterility with no sperm may be related to the expression abundance of *TSSK1B.* The histological analysis of cattle–yak testis also exhibited smaller seminiferous tubules with few SC, PS, and other cells. This finding was explained by other research on the cattle–yak; that is, such a phenomenon may be due to the germ-cell apoptosis being initiated at the pachytene stage and the meiosis being blocked [33]. This phenomenon was also observed in a medical case in which a patient with a heterozygous deletion in 5q22.2q23.1 (including *TSSK1B*) suffered from asthenoteratozoospermia, with cephalic defects, numerous atypical sperm morphologies, and low sperm motility, thus demonstrating that *TSSK1B* regulates spermatozoan development and quality [34]. Other studies on the transcriptome of gonad and gametogenesis demonstrated that the *TSSK1*-like gene is vital to male gonadal development and spermatogenesis [35,36]. Considering the specific expression of *TSSK1B* in yak testis with increasing age and the relationship between *TSSK1B* expression and the developmental status of the germline in testis, the low expression of *TSSK1B* in male cattle–yak testis may influence the generation of mature spermatozoa and lead to the infertility of male cattle–yak.

DNA methylation is an essential reversible epigenetic modification of the genome for mammalian gametogenesis and embryonic development, which undergoes extensive reprogramming during the oogenesis and spermatogenesis, with changes in the expression of related genes. DNA methylation has been proven to regulate gene expression and contribute to the alteration of the transcriptome and deregulation of cellular pathways [37]. It is usually considered to be a transcriptional repressive mark because of the ability to decrease the efficiency of promoters [38]. A study on the promoter region of yak *HIF* found that demethylation may be related to the higher expression of HIF in yak tissues than in general cattle [39]. Abnormal DNA methylation has always served as the biomarker of many diseases [40,41,42], such as cancers [43]. As for the genome, the upstream 2000 bp from the transcription start site is generally considered as the promoter region of the gene [25]. The CpG-rich regions, which are also known as CGIs, are widely recognized at the promoter region and associated with the TSS of genes [44]. In vertebrates, the inheritable DNA methylation often occurs at CpG sites and is related to the transcriptional silencing of genes [45]. The CGIs that overlap with the promoter regions usually remain unmethylated in the germline [46]. For further exploration of the potential mechanism of the expression suppression of *TSSK1B* in cattle–yak testis, the methylated CpG sites in the predicted CGIs of the *TSSK1B* promoter region were detected via BSP, a general method that is based on cloning and sequencing to detect DNA methylation at specific CpG sites, to contrast the methylation levels of *TSSK1B* in yak and cattle–yak testis. The results showed that the methylation level of the *TSSK1B* promoter region of cattle–yak was significantly higher than that of yak, in agreement with the result of RT-qPCR. This inhibition effect of the highly methylated promoter region was also reported in the study of sperm samples from infertile men, in which asthenoteratospermia and oligoasthenoteratospermia could easily appear and the methylation frequencies in the promoter regions of hsa-miR-449 and hsa-miR-34b family were significantly high [47,48]. Given the low expression of *TSSK1B* in cattle–yak testis, as shown in the RT-qPCR results, the higher methylation of CGIs in the *TSSK1B* promoter region of cattle–yak testis may downregulate the transcription and expression of *TSSK1B* and ultimately influence the spermateleosis in cattle–yak testis, which leads to infertility.

Throughout the whole study, we found a potential relationship between *TSSK1B* expression and male cattle–yak infertility. However, the complete mechanism and regulatory pathway of male infertility in male cattle–yak require further study, such as establishing cell and animal knockout models and conducting a proteome analysis.

## 5. Conclusions

In summary, *TSSK1B* was a relatively conserved gene that was specifically expressed in yak testis, especially after sexual maturity, indicating a close relationship to spermatogenesis. Furthermore, the differential expression of *TSSK1B* in the cattle–yak testis was associated with the hypermethylation of the promoter region, which may be one of the causes for the male sterility of cattle–yak.

## Figures and Tables

**Figure 1 animals-13-00320-f001:**
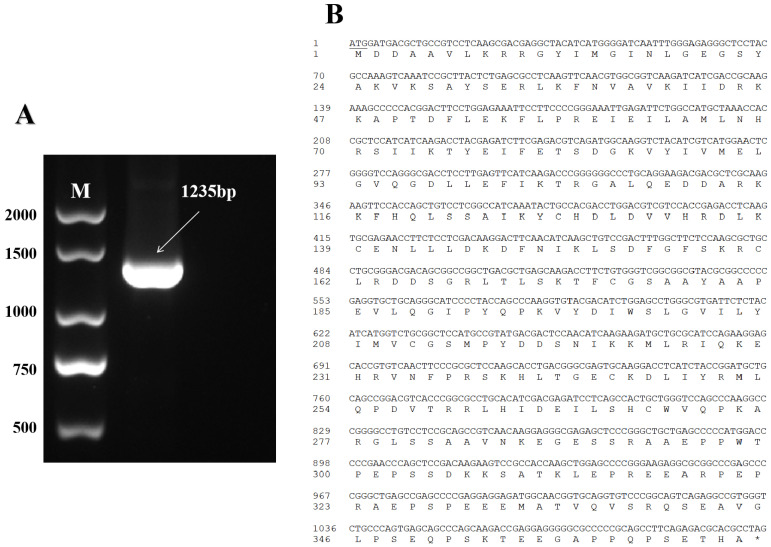
The sequence analysis of yak *TSSK1B* gene. (**A**) The agarose gel electrophoresis of *TSSK1B* amplification product obtained from adult yak testis. (**B**) The predicted coding protein sequence generated by ORF of yak *TSSK1B*. The underline represents the initiation codon, and the asterisk “*” denotes the termination codon.

**Figure 2 animals-13-00320-f002:**
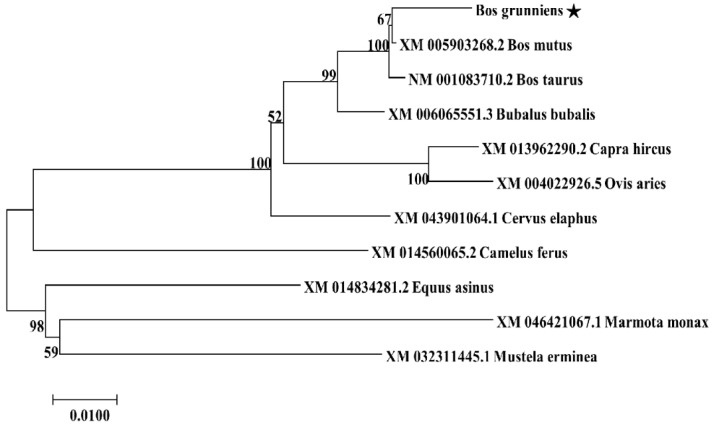
The phylogenetic tree of the *TSSK1B* gene established from the abovementioned alignment. The asterisk “★” represents the *TSSK1B* sequence obtained from yak testis.

**Figure 3 animals-13-00320-f003:**
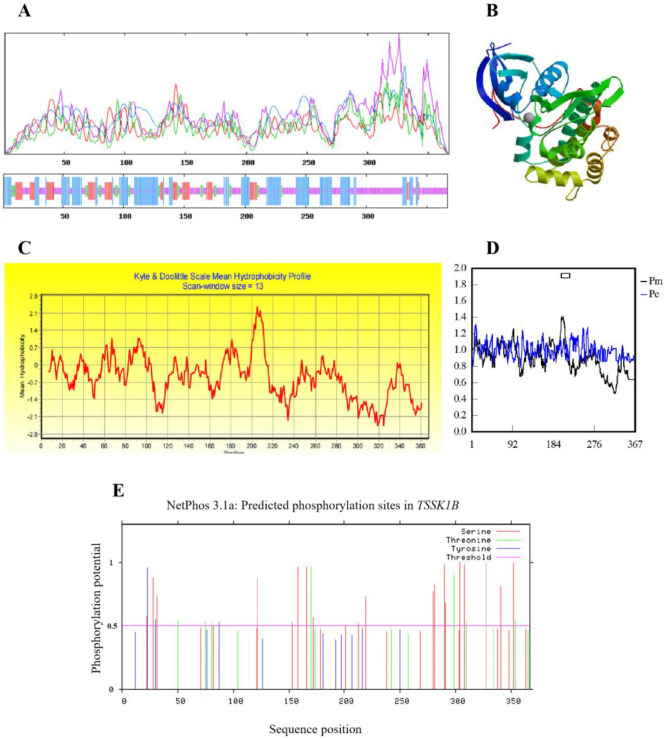
The physicochemical properties of predicted protein coding by *TSSK1B* gene. (**A**) The predicted secondary structure of protein encoded by *TSSK1B* gene. Blue line, alpha helix; green line, beta-turn; red line, extended strand; orange line, random coil. (**B**) The predicted tertiary structure of protein. (**C**) The predicted hydrophobicity profile of protein encoded by *TSSK1B* gene. Hydrophobic: The mean hydrophobicity of each position is more than 0. Hydrophilic: The mean hydrophobicity of each position is less than 0. (**D**) The transmembrane structure of protein coding by *TSSK1B* gene. The square frame represents the position transmembrane site. (**E**) The phosphorylation sites analysis of TSSK1B. The prediction scores above 0.5 indicate positive predictions.

**Figure 4 animals-13-00320-f004:**
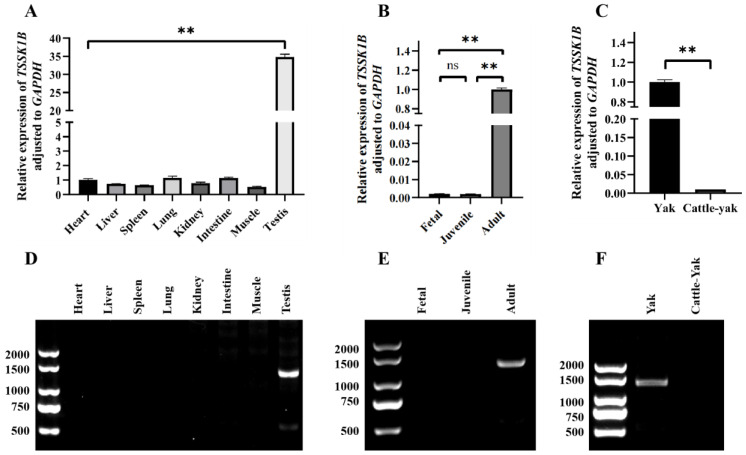
The expression pattern of *TSSK1B* gene in tissues of yak and cattle–yak based on the RT-PCR and RT-qPCR. (**A**) The expression level of *TSSK1B* gene in eight tissues of adult male yaks. (**B**) The expression level of *TSSK1B* gene in yak testis from different growth stages. (**C**) The expression level of *TSSK1B* gene in testis of adult yak and cattle–yak. (**D**) The amplification product of *TSSK1B* in adult male yak tissues detected by agarose gel electrophoresis. (**E**) The expression level of *TSSK1B* in different ages of yak testis, as detected by agarose gel electrophoresis. Analysis of *TSSK1B* expression by RT-PCR in different tissues. (**F**) The electrophoresis figure that reflected the expression level of *TSSK1B* between adult yak and cattle–yak testes. ** represents *p* < 0.01 and “ns” means there is no significance.

**Figure 5 animals-13-00320-f005:**
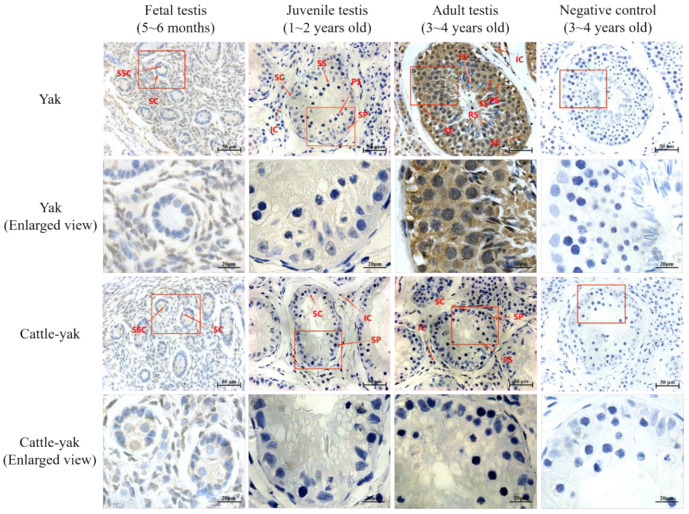
The localization and expression of *TSSK1B* in cattle–yak and yak testes. Testes from different stages (fetal, juvenile, and adult) of yak and cattle–yak development were embedded in paraffin, and sections of 4 μm thickness were prepared and stained with anti-*TSSK1B* antibody (brown) and counterstained with hematoxylin to visualize DNA (blue). The following pictures offer an enlarged view of the red box in the above figures. SSC, spermatogonial stem cells; SC, Sertoli cells; SP, spermatogonium; PS, primary spermatocyte; SS, secondary spermatocyte; IC, Leydig cells; RS, round spermatid; ES, elongating spermatid.

**Figure 6 animals-13-00320-f006:**
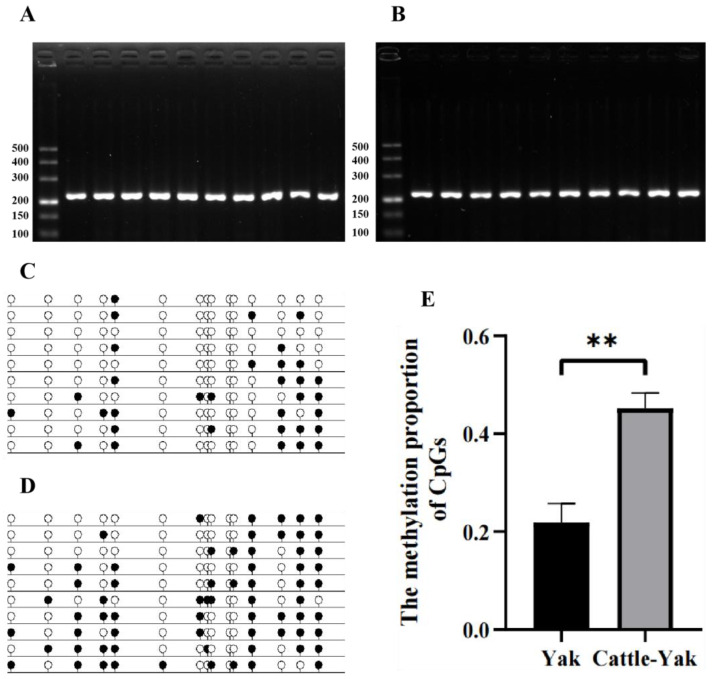
The methylation status of *TSSK1B* promoter region. (**A**,**B**) Ten BSP-amplified products of bisulfite-converted DNA sequence selected from *TSSK1B* promoter region, including the CpG island, in adult yak and cattle–yak testis, respectively. (**C**,**D**) The methylation status of CpG sites from adult yak and cattle–yak testes drew out with the sequencing results. The black circles represent methylated CpGs, and white circles represent unmethylated CpGs. (**E**) The methylation percentages of CpG sites in adult yak and cattle–yak testis calculated from (**C**,**D**). ** represents *p* < 0.01.

**Table 1 animals-13-00320-t001:** The primers for *TSSK1B* analysis.

Primer Name	Sequence (5′→3′)	Product Size (bp)	Annealing Temperature (°C)	Application
*pTSSK1B*	F: ATGTCCCGCAGGGATGTAGA R: CACGAAAATAGCGGCACGTC	1287	57 °C	RT-PCR
*qGAPDH*	F: GAAGGTCGGAGTGAACGGAT R: TGACTGTGCCGTTGAACTTG	170	60 °C	RT-qPCR
*qTSSK1B*	F: TGCACATCGACGAGATCCTCA R: AGCTCTCGCCCTCCTTGTTG	90	60 °C	RT-qPCR
*bTSSK1B*	F: TTTTTATTGGTTGATTAGAGAAGGG R: CTCAAAAATTCTCCAAATCTCCTC	231	57 °C	BSP

F, forward primer; R, reverse primer.

## Data Availability

Data sharing is not applicable to this article.

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
