# Peer review of "The Biological Characteristics and Differential Expression Patterns of TSSK1B Gene in Yak and Its Infertile Hybrid Offspring"

_animals, 2023, doi:10.3390/ani13020320_

Round 1

Reviewer 1 Report

In review to the submmitted work (The Biological characteristics and Diffrential expression pattern of TSSK1B gene in Yak and its infertile Hybrid Offspring).

The authors found TSSK1B gene is specifically expressed in Yak and highly expressed in adults. It rarely to be expressed in adult male yak and F1 generation. TSSK1B promoter is hypermethylated in adult cattle yak compared to yak. They concluded it might be related to male infertility.

The authors analyzed sequence of yak Tssk1b by electrophoresis, predicted the coding protein sequence, compared the ortholog sequences from NCBI databases with yak and designed phylogenetic tree. They did some analysis of the protein structure, hydrophobicity characters, phosphorylation sites. They measured expression patterns in tissues of yalk and cattle yak by RT and qRT PCR in addition to IHC, methylation status of protein  in these two species.

However the efforts that clearly shown here from the authors in this study, but the quality and the strength of represented work need to be higher to be published in animals journal. The conclusion that link the data to the male infertility still require more validation by more approaches not just detecting (descriptive) the expression levels in young or adult tissues, in yak or cattle yak.

I believe that at the current format of the manuscript I apologize for the authors that I could not recommend publication in animals Journal.

Author Response

Point: In review to the submmitted work (The Biological characteristics andDiffrential expression pattern of TSSK1B gene in Yak and its infertile HybridOffspring). The authors found TSSK1B gene is specifically expressed in Yak and highlyexpressed in adults. It rarely to be expressed in adult male yak and F1 generation. TSSK1B promoter is hypermethylated in adult cattle yak compared to yak. Theyconcluded it might be related to male infertility. The authors analyzed sequence of yak Tssk1b by electrophoresis, predicted the codingprotein sequence, compared the ortholog sequences from NCBI databases withyakand designed phylogenetic tree. They did some analysis of the protein structure, hydrophobicity characters, phosphorylation sites. They measured expression patterns in tissues of yalk and cattle yak by RT and qRT PCR in addition to IHC, methylationstatus of protein in these two species. However the efforts that clearly shown here from the authors in this study, but thequality and the strength of represented work need to be higher to be publishedinanimals journal. The conclusion that link the data to the male infertility still requiremore validation by more approaches not just detecting (descriptive) the expressionlevels in young or adult tissues, in yak or cattle yak. I believe that at the current format of the manuscript I apologize for the authors that I could not recommend publication in animals Journal.

Response: Thank you for your valuable comments and suggestions. With your help, we believe we could do better in the future. Although the content of this manuscript is not very rich, it has certain scientific guiding significance and value for the analysis of male cattle-yak sterility. As we all known, yak is one of the specific species which can adapt to the alpine and hypoxia environment and play a crucial role in local livelihood and economy. In order to solve the low production of yak, cattle-yak is an effective method to improve the production efficiency. However, male cattle-yak is infertile as the spermatogenesis is blocked[Ref.1-5]. The accumulating evidences indicate that TSSK1B plays key role in spermatogenesis and male fertility [Ref.6-8]. Therefore, we hypothesized that male cattle-yak sterility is related to the abnormal expression of TSSK1B gene. In this study, we first cloned the sequence of TSSK1B in yak, and compared the biological characteristics and expression pattern between yak and catlle-yak. Consistent with expectations, the mRNA expression level of TSSK1B is significantly lower in cattle-yak. Further analysis found that the promoter region of TSSK1Bwas hypermethylation in testis of cattle-yak. Thus, we concluded the low expression of TSSK1B in cattle-yak is associated with the hypermethylation, which might be lead to male sterility. Of course, the above results are not enough to prove that the male sterility of cattle-yak is caused by TSSK gene. More work needs to be carried out, such as establishing animal models and cell models. This is also our ongoing work, and we look forward to sharing it with you soon.

[1] Yang L, Min X, Zhu Y, et al. Polymorphisms of SORBS 1 Gene and Their Correlation with Milk Fat Traits of Cattleyak[J]. Animals, 2021, 11(12): 3461.

[2] Xu C, Wu S, Zhao W, Mipam T, Liu J, Liu W, et al. Differentially expressed microRNAs between cattleyak andyak testis. Sci Rep 2018; 8(1): 592.

[3] Yu S, Cai X, Sun L, Zuo Z, Mipam T, Cao S, et al. Comparative iTRAQ proteomics revealed proteins associatedwith spermatogenic arrest of cattleyak. J Proteomics 2016: 142: 102–13.

[4] Cai X, Yu S, Mipam T, Yang F, Zhao W, Liu W, et al. Comparative analysis of testis transcriptomes associatedwith male infertility in cattleyak. Theriogenology 2017; 88: 28–42.

[5] Sun L, Mipam TD, Zhao F, Liu W, Zhao W, Wu S, et al. Comparative testis proteome of cattleyak fromdifferent developmental stages. Animal 2017; 11(1): 101–11.

[6] Xu, B., Hao, Z., Jha, K. N., Zhang, Z., Urekar, C., Digilio, L., ... & Herr, J. C. (2008). Targeted deletion of Tssk1 and2 causes male infertility due to haploinsufficiency. Developmental biology, 319(2), 211-222.

[7] Salicioni, A. M., Gervasi, M. G., Sosnik, J., Tourzani, D. A., Nayyab, S., Caraballo, D. A., & Visconti, P. E. (2020). Testis-specific serine kinase protein family in male fertility and as targets for non-hormonal male contraception. Biology of Reproduction, 103(2), 264-274.

[8] Omolaoye, T. S., Hachim, M. Y., & du Plessis, S. S. (2022). Using publicly available transcriptomic data to identify mechanistic and diagnostic biomarkers in azoospermia and overall male infertility. Scientific Reports, 12(1), 1-17.

Reviewer 2 Report

In this manuscript, Zhu et al. investigate the gene TSSK1B and its expression in yak and cattle-yak. The sequence of TSSK1B was compared among species and its RNA was detected and compared among different organs. It is very interesting to find TSSK1B is highly expressed in testis but not in the cattle-yak. A few questions to help readers understand this gene well as below. 

1. Have authors detected the protein level of TSSK1B?

2. Have authors tried to detect its expression in sperm? Do authors think the TSSK1B is specific to sperm? 

3. Is a TSSK1B knockout model available? What is the function of TSSK1B?

Author Response

Response to Reviewer 2 Comments
Point 1: In this manuscript, Zhu et al. investigate the gene TSSK1B and its expression in yak and cattle-yak. The sequence of TSSK1B was compared among species and its RNA was detected and compared among different organs. It is very interesting to find TSSK1B is highly expressed in testis but not in the cattle-yak. A few questions to help readers understand this gene well as below.

Response 1: Thanks for your encouragement and comments, which help to improve our manuscript and further study.

Point 2: Have authors detected the protein level of TSSK1B?
Response 2: Thanks for your suggestion. In this study, due to the limited by
laboratory conditions, we did not detect the protein level of TSSK1B by western blot, which is truly a necessary index to be conducted. However, the IHC was instead of WB to quantify, on a certain degree, the protein expression of TSSK1B in testis of yak and cattle-yak. In the future study, we will improve our experimental conditions and analyze its expression from multiple perspectives.

Point 3: Have authors tried to detect its expression in sperm? Do authors think the TSSK1B is specific to sperm?

Response 3: Thanks for your suggestion. According our IHC results, the strong
positive signal of TSSK1B were detected in the adult yak testis, especially in different types of germ cells, which was consist with the previous research in mouse and human that pointed out the main expression of TSSK1B in late spermiogenesis and condensed spermatids (Li et al., 2011, showed in following figure). However, there is nearly no sperm in testis of cattle-yak (also shown in Figure 5), as the blocked spermatogenesis. Therefore, it is impossible to study the expression of TSSK1B in the sperm of cattle-yak. Certainly, we are preparing to compare the expression in sperm of cattle and yak, which is benefit for exploring the effects of TSSK1B on male infertile of cattle-yak.
Figure [Li et al., 2011]. Immunolocalization of TSSK1 in mouse and human sperm. Localization in mouse sperm (above), and human sperm (below). Li Y, Sosnik J, Brassard L, et al. Expression and localization of five members of the testis-specific serine kinase (Tssk) family in mouse and human sperm and testis[J]. Molecular Human Reproduction, 2011, 17(1): 42-56.

Point 4: Is a TSSK1B knockout model available? What is the function of TSSK1B?
Response 4: Thanks for your advice and question. The TSSK1B knockout model have been mentioned in previous research in mouse (Xu et al., 2008), and TSSK1/2 double KO model was also established (Shang et al., 2010). At present, no relevant report on large animals. According to the previous study and our discovery, TSSK1B is known to play a major role in signaling events associated with sperm differentiation and function. The model of TSSK1/2 double KO found that the male infertility phenotype resulting from lack of expression of Tssk1/Tssk2 was a consequence of developmental dysregulation
and the collapsing of the mitochondrial sheet in late spermatids. Mechanistically, Tssk1/2 are involved in the functional transformation of the chromatoid body, known to be involved in RNA storage and metabolism and is normally located in the cytoplasm of male germ cells. Shang P, Baarends WM, Hoogerbrugge J, Ooms MP, van Cappellen WA, de Jong AA, Dohle GR, van Eenennaam H, Gossen JA, Grootegoed JA. Functional transformation of the chromatoid body in mouse spermatids requires testis-specific serine/threonine kinases. J Cell Sci 2010; 123:331–339

Reviewer 3 Report

In the study, the authors studied the expression of TSSK1B gene in both yaks and cattle-yak hybrids, and found that TSSK1B highly expressed in the testis of adult yaks and TSSK1B gene in adult cattle-yak testis was hypermethylated. The findings of the study facilitate the further study in this field.

This is an interesting study. However, I have following comments to the authors.

1.       For the TSSK1B gene expressed in cattle-yak testis, what is it like? Is it cattle TSSK1B expressing in the cattle-yak hybrids or yak TSSK1B or both of them?  It is important to know that, though there is high similarity between them. It is related with the expressing detection in various tissues.

2.       The authors should provide information about whether there is blood relationship among the studied individuals.

3.       The English writing should be carefully checked and polished.

4.       The reference number in the main text of the manuscript should be revised and properly demonstrated.

Author Response

Thank you for your efforts on our manuscript. All the comments you mentioned have been replied, please see the attachment.

Point 1: In the study, the authors studied the expression of TSSK1B gene in both yaks and cattle-yak hybrids, and found that TSSK1B highly expressed in the testis of adult yaks and TSSK1B gene in adult cattle-yak testis was hypermethylated. The findings of the study facilitate the further study in this field. This is an interesting study. However, I have following comments to the authors.

Response 1: Thank you for your encourage and suggestion, which are constructive to our manuscript and our future research.

Point 2: For the TSSK1B gene expressed in cattle-yak testis, what is it like? Is it cattleTSSK1B expressing in the cattle-yak hybrids or yak TSSK1B or both of them? It is important to know that, though there is high similarity between them. It is related with the expressing detection in various tissues.

Response 2: Thanks for your suggestion. In this study, we cloned the sequence of yak TSSK1B gene, and compared with the homology of other species, especially with cattle. The result showed the similarity is 99.42% to Bos taurus, only a few mutations in CDS region without amino acid change. So, it’s difficult to distinguish the expression of TSSK1B in cattle-yak from cattle or yak. Furthermore, cattle-yak was only produced by artificial insemination (AI) with commercialized frozen bovine semen, because cattle cannot survive in these plateau environments. Thus, it is difficult to trace the paternal sample, only the maternal individual can be found. The expression of TSSK1B during the development of male yak was detected in this study (Fig.4B), and found that TSSK1B was highly expressed in adult yak testis, which suggest TSSK1B is associated with sexual maturation and spermatogenesis. On the contrary, the mRNA expression of TSSK1B in cattle-yaktestis is hardly expressed, and it is related to the biological phenomenon of azoospermia. In addition, the tissue expression profile showed that TSSK1B gene was specifically expressed in yak testis (Fig.4A), and no expression in any tissue of cattle-yak (data not show).

Point 3: The authors should provide information about whether there is blood relationship among the studied individuals.

Response 3: Thanks for your suggestion. In this study, the samples of yak and cattle-yak were collected from the same area but different pasture. So, there is almost no blood relationship among them, although no relevant verification. This is very important for this study and has a certain impact on the results. In next, we will establish a yak, cattle, and cattle-yak model with known blood relationship to verify this study and further explore relevant mechanisms.

Point 4: The English writing should be carefully checked and polished.

Response 4: Thanks for your suggestion. We have strictly revised the language issue and with the help of EnPapers, a company dedicated to helping international researchers publish their findings in the best English language journals, to check the manuscript carefully.

Point 5: The reference number in the main text of the manuscript should be revised and properly demonstrated.

Response 5: Thanks for your advice. The formation of reference in our main text has been carefully checked and revised

Round 2

Reviewer 1 Report

I would you like to thank the authors for their reply to my comments. Regarding to their reply, it would be better to highlight in the last part of the discussion and in the conclusion the following statement "that role of our results suggest a possible relation with male infertility in the Yalk and this will require a further research"

Author Response

Point: I would you like to thank the authors for their reply to my comments. Regarding to their reply, it would be better to highlight in the last part of the
discussion and in the conclusion the following statement "that role of our results
suggest a possible relation with male infertility in the Yak and this will require a
further research".

Response: Thank you for the suggestion. We have already revised this issue in the manuscript by clarifying the contribution, limitation and expectation of our research at the end of discussion to make the content clearer.

Reviewer 2 Report

The reviewer thanks for authors' reply. Unfortunately, no obvious revision has been done according to reviewer's comments which could have improved the interpretation of the work. 

Author Response

Thank you for your efforts on our manuscript. The comments you mentioned have been replied, please see the attachment.

Point: The reviewer thanks for authors' reply. Unfortunately, no obvious revision has been done according to reviewer's comments which could have improved the interpretation of the work.

Response: Thank you for your comment, which are very constructive to our ongoing work. Of course, the current results are not enough to completely prove that the male sterility of cattle yak is caused by the abnormal of TSSK1B, but it truly points out remarkable clue that the low expression of TSSK1B might be related to the male cattle-yak infertility, which provide a theoretical basis for further study on sterility of male cattle-yak.
Given the absence of sperms in adult male cattle-yak testis, it is hard to detect any protein expression in mature cattle-yak sperm. However, based on our current study on temporal expression of TSSK1B in yak and cattle-yak testis, it is possible to ulteriorly study on the early germ cell such as spermatogonial stem cells to explore if there is a prophase regulation mechanism of TSSK1B in testis.
